# UIP2P: UNSUPERVISED INSTRUCTION-BASED IMAGE EDITING VIA CYCLE EDIT CONSISTENCY

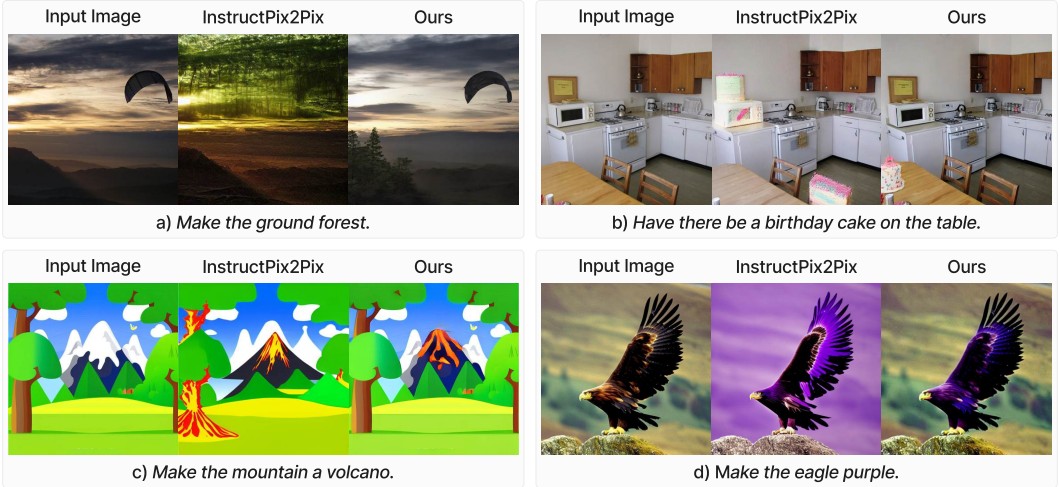

Figure 1: **Unsupervised InstructPix2Pix.** Our model achieves more precise and coherent edits while preserving the original structure of the scene via Cycle Edit Consistency. UIP2P outperforms InstructPix2Pix in both real images—(a) and (b)—and synthetic images—(c) and (d).

## ABSTRACT

We propose an unsupervised model for instruction-based image editing that eliminates the need for ground-truth edited images during training. Traditional supervised approaches depend on datasets containing triplets of input image, edited image, and edit instruction, often generated by either existing editing methods or human-annotations, which introduce biases and limit their generalization ability. Our model addresses these challenges by introducing a novel editing mechanism called Cycle Edit Consistency (CEC). We propose to apply a forward and backward edit in one training step and enforce consistency in both the image and attention space. This allows us to bypass the need for ground-truth edited images and unlock training on datasets comprising either real image-caption pairs or image-caption-edit triplets. We empirically show that our unsupervised method achieves better performance across a wider range of edits with high fidelity and precision. By eliminating the need for pre-existing datasets of triplets, reducing biases associated with supervised methods, and introducing CEC, our work represents a significant advancement in unblocking scaling of instruction-based image editing.

## 1 INTRODUCTION

Diffusion models (DMs) have recently achieved significant advancements in generating high-quality and diverse images, primarily through breakthroughs in text-to-image generation (Ho et al., 2020; Saharia et al., 2022; Rombach et al., 2022; Ramesh et al., 2022). This led to the development of various techniques for tasks like personalized image generation (Ruiz et al., 2023; Wei et al., 2023; Gal et al., 2022a), context-aware inpainting (Lugmayr et al., 2022; Nichol et al., 2022; Yang et al.,

Figure 2: **Examples of biases introduced by Prompt-to-Prompt in the InstructPix2Pix dataset.** Each example shows an input image and its corresponding edited image (generated by Prompt-to-Prompt) along with the associated edit instruction. *(a) Attribute-entangled edits*: modifying the lady's dress also unintentionally changes the background. *(b) Scene-entangled edits:* transforming the cottage into a castle affects surrounding elements. *(c) Global scene changes:* converting the image to black and white alters the entire scene.

2023), and editing images based on textual prompts (Avrahami et al., 2022; Hertz et al., 2022; Meng et al., 2022; Kawar et al., 2022; Couairon et al., 2023). Editing images based on textual instructions (Brooks et al., 2023) demonstrates the versatility of DMs as robust tools for a wide array of image editing tasks.

However, existing methods predominantly rely on supervised learning, which requires large datasets of triplets containing input and edited images with edit instructions (Brooks et al., 2023; Zhang et al., 2023a;b; Fu et al., 2023). These datasets are often generated using editing methods such as Prompt-to-Prompt (Hertz et al., 2022) or human annotations. While the former solution allows better scaling of dataset size, it, unfortunately, introduces biases, such as (a) attribute-entangled or (b) scene-entangled edits that affect unintended parts of the image or (c) cause significant changes to the entire scene (see Fig. 2). On the other hand, human-annotated data, though valuable, is impractical for large-scale training due to the high cost and effort involved in manual annotation. This reliance on human-annotated or generated ground-truth edited images limits the diversity of the achievable edits. It hinders the development of models capable of understanding and executing a wide range of user instructions.

We present UIP2P, an unsupervised model for instruction-based image editing that removes the dependency on datasets of triplets, generated or human-annotated, by introducing Cycle Edit Consistency (CEC), *i.e.*, consistency obtained by applying forward and reverse edits. Leveraging the alignment between text and images in the CLIP embedding space (Radford et al., 2021b), CEC ensures that edits remain consistent. By enforcing consistency in both the image and attention space, the UIP2P model accurately interprets and localizes user instructions, ensuring that edits are coherent and reflect the intended changes throughout the entire editing process. CEC allows UIP2P to effectively maintain the integrity of the original content while making precise modifications, further enhancing the reliability of the edits. This approach unlocks the ability to train on large real-image datasets by eliminating the need for pre-existing datasets. As a result, our approach significantly broadens the scope and scalability of instruction-based image editing compared to previous methods.

Our key contributions are as follows:

- We introduce an unsupervised model for instruction-based image editing, UIP2P, that removes the requirement for ground-truth edited images during training, providing a more scalable and adaptable alternative to current supervised methods.

- We introduce Cycle Edit Consistency (CEC), a novel method that ensures consistent edits when cycled across forward and reverse editing, maintaining coherence in both the image and attention space. This allows precise, high-fidelity edits that accurately reflect user instructions.

- Our approach demonstrates scalability and versatility across various real-image datasets, enabling a wide range of edits without relying on pre-existing datasets, significantly broadening the scope of instruction-based image editing.

## 2 RELATED WORK

**CLIP-Based Image Manipulation.** Patashnik et al. (2021) introduces StyleCLIP, which combines StyleGAN and CLIP for text-driven image manipulation, requiring optimization for each specific edit. Similarly, Gal et al. (2022b) presents StyleGAN-NADA, enabling zero-shot domain adaptation by using CLIP guidance to modify generative models. While these approaches allow for flexible edits, they often rely on domain-specific models or optimization processes for each new task. These works illustrate the potential of CLIP's powerful semantic alignment for image manipulation, which motivates the use of CLIP in other generative frameworks, such as diffusion models.

**Text-Driven Image Editing with Diffusion Models.** One common approach in text-driven image editing is to use pre-trained diffusion models by first inverting the input image into a latent space and then applying edits through text prompts (Mokady et al., 2022; Hertz et al., 2022; Wang et al., 2023b; Meng et al., 2022; Couairon et al., 2023; Ju et al., 2023; Parmar et al., 2023; Wang et al., 2023a; Wu et al., 2023). For example, DirectInversion (Ju et al., 2023) edits the image after inversion using Prompt-to-Prompt (Hertz et al., 2022), but this can lead to losing essential details from the original image. In addition, methods like CycleDiffusion (Wu & la Torre, 2023), CycleNet (Xu et al., 2023), and DualDiffusion (Su et al., 2022) explore domain-to-domain translation as a way to improve image editing. However, their focus on translating between two fixed domains makes it difficult to handle more complex edits, such as the insertion or deletion of objects. In contrast, we focus on a general-purpose image editing approach that is not limited to domain translation, enabling greater flexibility in handling a wider variety of edits.

Another line of methods for image editing involves training models on datasets containing triplets of input image, edit instruction, and edited image such as InstructPix2Pix (Brooks et al., 2023; Zhang et al., 2023a;b). These methods, since they directly take the input image as a condition, do not require an inversion step. InstructDiffusion (Geng et al., 2023) builds on InstructPix2Pix by handling a wider range of vision tasks but has difficulty with more advanced reasoning. MGIE (Fu et al., 2023) improves on this by using large multimodal language models to generate more precise instructions. SmartEdit (Huang et al., 2024) goes a step further by introducing a Bidirectional Interaction Module that better connects the image and text features, improving its performance in challenging editing scenarios.

A significant challenge in image editing is the lack of large-scale triplet datasets. Instruct-Pix2Pix (Brooks et al., 2023) addresses this by generating a large dataset using GPT-3 (Brown et al., 2020) and Prompt-to-Prompt (Hertz et al., 2022). However, while this solves the data scarcity issue, it introduces new challenges, such as model biases stemming from Prompt-to-Prompt. MagicBrush (Zhang et al., 2023a) attempts to overcome this with manually annotated datasets, but this approach is small-scale and impractical for broader use.

Our method leverages CLIP's semantic space, which aligns image and text, to offer a more robust solution. It addresses both the dataset limitation and model bias problems by introducing Cycle Edit Consistency (CEC), which ensures consistency across forward and reverse edits. This approach not only improves scalability and precision for handling complex instructions but also eliminates the need for triplet datasets, making it compatible with any image-caption dataset of real images. Furthermore, since CEC modifies only the training phase of InstructPix2Pix, it can be seamlessly integrated with any extension of the model.

## 3 BACKGROUND

### 3.1 LATENT DIFFUSION MODELS (LDMS)

Stable Diffusion (SD) is a prominent Latent Diffusion Model (LDM) designed for text-guided image generation (Rombach et al., 2022). LDMs operate in a compressed latent space, typically derived from the bottleneck of a pre-trained variational autoencoder, which enhances computational efficiency. Starting with Gaussian noise, the model progressively constructs images through an iterative denoising process guided by text conditioning. This process is powered by a U-Net-based architecture (Dhariwal & Nichol, 2021), utilizing self-attention and cross-attention layers. Self-attention helps refine the evolving image representation, while cross-attention integrates the textual guidance.

Cross-attention mechanisms are crucial in directing image generation in LDMs. Each cross-attention layer consists of three main components: queries (Q), keys (K), and values (V). Queries are generated from intermediate image features through a linear transformation ($f_Q$), while keys and values are extracted from the text conditioning using linear transformations ($f_K$ and $f_V$). The attention mechanism, formulated in Eq. (1), computes attention maps that indicate which regions of the evolving image should be modified based on the text description. We utilize these maps in our loss functions to regulate and localize the desired edit, enabling localized and consistent image editing.

$$\text{Attention}(Q, K, V) = \text{Softmax}\left(\frac{QK^T}{\sqrt{d}}\right) \cdot V \tag{1}$$

## 3.2 INSTRUCTPIX2PIX (IP2P)

Our method builds upon InstructPix2Pix (IP2P) (Brooks et al., 2023), an LDM-based framework for text-conditioned image-to-image transformations. Like Stable Diffusion, IP2P employs a U-Net architecture. The conditional framework of IP2P allows it to simultaneously utilize both an input image ($I$) and a text instruction ($T$) to guide image modifications. Classifier-free guidance (CFG) (Ho & Salimans, 2021) is used, with coefficients ($s_I$ and $s_T$) controlling the influence of the text and the original image during editing. The predicted noise vectors ($e_\theta$) from the learned network are combined linearly, as described in Eq. (2), to produce the final score estimate $\tilde{e}_\theta$.

$$\begin{aligned}
\tilde{e}_\theta(z_t, c_I, c_T) = {}& e_\theta(z_t, \varnothing, \varnothing) \\
& + s_I \cdot (e_\theta(z_t, c_I, \varnothing) - e_\theta(z_t, \varnothing, \varnothing)) \\
& + s_T \cdot (e_\theta(z_t, c_I, c_T) - e_\theta(z_t, c_I, \varnothing)).
\end{aligned} \tag{2}$$

InstructPix2Pix is trained on a dataset containing triples of input image, edit instruction and edited image. The dataset is composed of synthetic images generate by SD on top of real captions, edit instructions generated by an LLM and edited images obtained using Prompt-to-Prompt (Hertz et al., 2022). The reliance on synthetic datasets introduces several limitations that we aim to address in this work. First, models like IP2P are trained exclusively on synthetic data, which limits their applicability during training on real-world image datasets. Second, their performance is inherently constrained by the quality of the images generated by Prompt-to-Prompt methods, which introduces biases into the editing process, as demonstrated in Fig. 2.

## 4 METHOD

Differently from related work such as InstructPix2Pix (Brooks et al., 2023), which rely on paired datasets of input and edited images for instruction-based editing, we utilize instead an unsupervised framework that requires only real images and corresponding edit instructions, eliminating the need for ground-truth edited images. In a nutshell, given an image and a forward edit instruction (*e.g.*, "turn the sky pink"), we generate an edited image. We then apply a reverse instruction (*e.g.*, "turn the sky blue.") to the edited image, aiming to recover the original input. During this forward-reverse edits, we enforce our proposed Cycle Edit Consistency (CEC) ensuring that the edits are reversible and maintain coherence in both the image and attention space. This approach allows us to scale instruction-based image editing across various real-image datasets without the limitations of synthetic or paired edited datasets. In the following sections, we describe our approach in detail, including the key components of our framework (Sec. 4.1), the loss functions used to enforce consistency, and the training data generation procedure (Sec. 4.2).

### 4.1 FRAMEWORK

#### 4.1.1 UIP2P

At the core of our method is the concept of Cycle Edit Consistency (CEC), which ensures that edits applied to an image can be reversed back to the original input through corresponding reverse

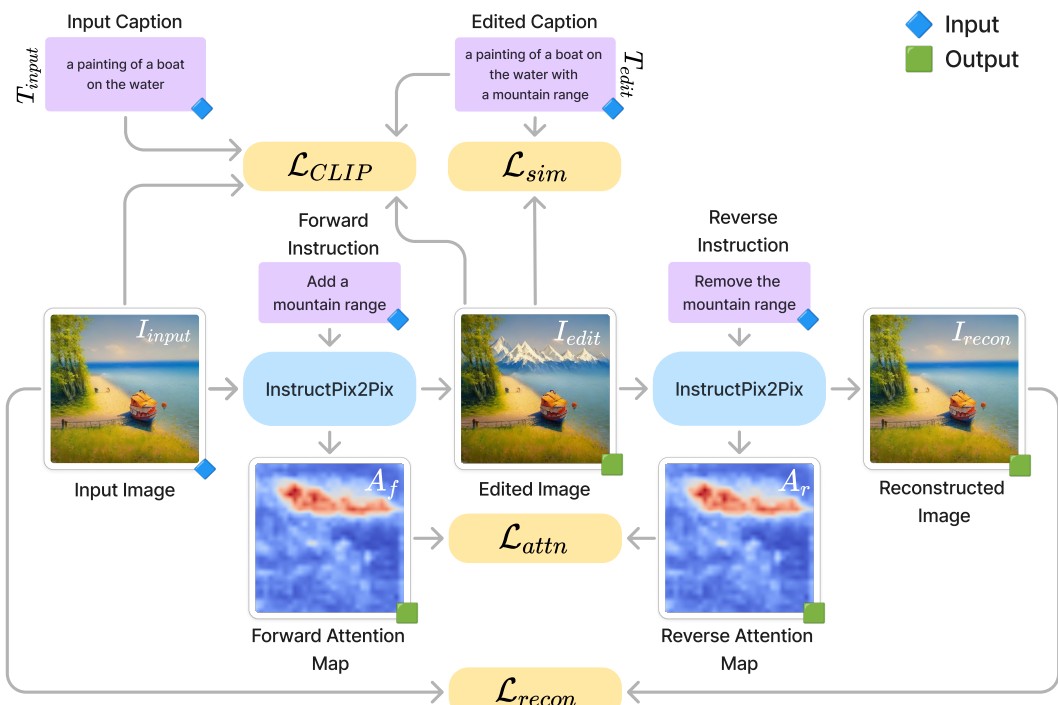

Figure 3: **Overview of the UIP2P framework.** The model performs instruction-based image editing by utilizing forward and reverse instructions. Starting with an input image and a forward instruction, the model generates an edited image using InstructPix2Pix. A reverse instruction is then applied to reconstruct the original image, enforcing Cycle Edit Consistency (CEC).

instructions. Our framework, UIP2P, introduces four key components designed to enforce CEC and maintain both semantic and visual consistency during the editing process, leveraging a mechanism that effectively reuses predictions across diffusion steps to enhance the editing process (an overview is illustrated in Fig. 3):

1. **Text and Image Direction Consistency**: We leverage CLIP embeddings (Radford et al., 2021a) to align the semantic relationship between textual instructions and the image modifications. By operating within CLIP's embedding space, our model ensures that the relationship between the input and edited images corresponds to the relationship between their respective captions. This alignment is critical for enforcing Cycle Edit Consistency (CEC), ensuring that the desired edit is applied while preserving the input image's structure.

2. **Attention Map Consistency**: To maintain consistency throughout the editing process, we enforce that attention maps generated during both forward and reverse edits align. This guarantees that the model consistently focuses on the same regions of the image during the initial edit and its reversal. Attention Map Consistency is crucial for CEC, as it ensures that localized edits can be effectively reversed.

3. **Reconstruction Consistency**: Central to enforcing CEC, the model must reconstruct the original input image after applying the reverse instruction. This ensures that the model can reliably undo its edits. We achieve this by minimizing both pixel-wise and semantic discrepancies between the reconstructed image and the original input, ensuring coherence between the applied edit and its reversal.

4. **Unified Prediction with Varying Diffusion Steps**: We sample different diffusion steps ($t$ for forward and $\hat{t}$ for reverse). Then, we independently predict $\hat{\epsilon}_F$ and $\hat{\epsilon}_R$ for one step of each, then apply them across $t$ steps in the forward (F) and $\hat{t}$ steps in the reverse (R) to reconstruct the image. Reusing the same prediction across steps reduces computational cost. By working with two similar images at different noise levels, the model learns to align its predictions, improving efficiency and ensuring more accurate edits.

By combining these components—Text and Image Direction Consistency, Attention Map Consistency, Reconstruction Consistency, and Unified Prediction with Varying Diffusion Steps—our framework not only enforces CEC, but also effectively applies across diverse real-image datasets. Importantly, this applicability is achieved without requiring ground-truth edited images, making the framework applicable to a wide range of tasks where annotated data is limited or unavailable. This ability to generalize beyond synthetic datasets underscores the versatility of our method in real-world instruction-based image editing scenarios.

### 4.1.2 Loss Functions

To enforce CEC and ensure both visual and semantic consistency during the editing and reconstruction process, we introduce the following loss terms:

**CLIP Direction Loss**: This loss ensures that the transformations applied to the image align with the text instructions in CLIP's semantic space (Gal et al., 2022b). Given the CLIP embeddings of the input image ($E_{I_{\text{input}}}$), edited image ($E_{I_{\text{edit}}}$), input caption ($E_{T_{\text{input}}}$), and edited caption ($E_{T_{\text{edit}}}$), the loss is defined as:

$$\mathcal{L}_{\text{CLIP}} = 1 - \cos\left(E_{I_{\text{edit}}} - E_{I_{\text{input}}},\ E_{T_{\text{edit}}} - E_{T_{\text{input}}}\right) \tag{3}$$

This loss aligns the direction of change in the image space with the direction of the transformation described in the text space, ensuring that the modifications reflect the intended edits and enabling reversible transformations necessary for CEC. This ensures that the model aligns transformations in the image space with the corresponding text modifications. However, ensuring spatial consistency is equally important, which we address with the Attention Map Consistency Loss (see next).

**Attention Map Consistency Loss**: To ensure that the same regions of the image are edited in both the forward and reverse edits, we define an attention map consistency loss. Let $A_f$ and $A_r$ represent the cross-attention maps from the forward and reverse edits, respectively. The loss is defined as:

$$\mathcal{L}_{\text{attn}} = \sum_i \left\| A_f^{(i)} - A_r^{(i)} \right\|_2 \tag{4}$$

where $i$ indexes the layers of the U-Net model. This loss ensures spatial consistency during both the editing and reversal stages, a key requirement for CEC, as it guarantees that the attention focuses on the same regions when reversing the edits.

**CLIP Similarity Loss**: This loss encourages the edited image to remain semantically aligned with the provided textual instruction. It is calculated as the cosine similarity between the CLIP embeddings of the edited image ($E_{I_{\text{edit}}}$) and the edited caption ($E_{T_{\text{edit}}}$):

$$\mathcal{L}_{\text{sim}} = 1 - \cos(E_{I_{\text{edit}}}, E_{T_{\text{edit}}}) \tag{5}$$

This loss ensures that the generated image aligns with the desired edits in the instruction, preserving semantic coherence between the forward and reverse processes—an essential aspect of CEC.

**Reconstruction Loss**: To guarantee that the original image is recovered after the reverse edit, we employ a reconstruction loss. This loss consists of two components: a pixel-wise loss and a CLIP-based semantic loss. The total reconstruction loss is defined as:

$$\mathcal{L}_{\text{recon}} = \|I_{\text{input}} - I_{\text{recon}}\|_2 + 1 - \cos(E_{I_{\text{input}}}, E_{I_{\text{recon}}}) \tag{6}$$

This loss ensures that the model can faithfully reverse edits and return to the original image when the reverse instruction is applied, enforcing CEC by minimizing differences between the input and reconstructed images.

### 4.1.3 Total Loss

The total loss function, is applied to the single step noise prediction rather than recursively, used to train the model is a weighted combination of the individual losses:

$$\mathcal{L}_{\text{CEC}} = \lambda_{\text{CLIP}}\mathcal{L}_{\text{CLIP}} + \lambda_{\text{attn}}\mathcal{L}_{\text{attn}} + \lambda_{\text{sim}}\mathcal{L}_{\text{sim}} + \lambda_{\text{recon}}\mathcal{L}_{\text{recon}} \tag{7}$$

where $\lambda_{\text{CLIP}}$, $\lambda_{\text{attn}}$, $\lambda_{\text{sim}}$, and $\lambda_{\text{recon}}$ are hyperparameters controlling the relative weights of each loss.

## 4.2 TRAINING DATA

To facilitate CEC training on datasets with image and edit instructions Brooks et al. (2023), we leverage Large Language Models (LLMs), such as GEMMA2 (Team et al., 2024) and GEMINI (Team et al., 2023), to automatically generate reverse edit instructions. These LLMs provide an efficient and scalable solution for obtaining reverse instructions with minimal cost and effort (Brooks et al., 2023). We use GEMINI Pro to enrich our dataset with reverse instructions based on the input caption, edit instruction, and corresponding edited caption. To improve model performance, we employ few-shot prompting during this process, enabling the generation of reverse instructions without the need for manually paired datasets, which significantly enhances scalability. The reverse instructions generated by the LLM aim to revert the edited image to its original form (see Tab. 1 - IP2P section).

Table 1: **Reverse Instruction Generation.** Our method generates reverse instructions for the IP2P dataset, eliminating the need for manually edited images. Additionally, edit instructions, edited captions, and reverse instructions are generated for CC3M and CC12M datasets—denoted as CCXM. The texts are generated by LLMs such as GEMINI Pro, and GEMMA2.

|  | Input Caption | Edit Instruction | Edited Caption | Reverse Instruction |
|---|---|---|---|---|
| **IP2P** | A man wearing a denim jacket | make the jacket a rain coat | A man wearing a rain coat | make the coat a denim jacket |
|  | A sofa in the living room | add pillows | A sofa in the living room with pillows | remove the pillows |
|  | . . . | . . . | . . . | . . . |
| **CCXM** | Person on the cover of a magazine | make the person a cat | Cat on the cover of the magazine | make the cat a person |
|  | A tourist rests against a concrete wall | give him a backpack | A tourist with a backpack rests against a concrete wall | remove his backpack |
|  | . . . | . . . | . . . | . . . |

Using the enriched dataset with reverse instructions (see Tab. 1, IP2P section), we fine-tune GEMMA2 (Team et al., 2024), to generate an edit instruction, edited caption, and reverse instruction given an input caption. We use this fine-tuned model to allow training on image-caption paired datasets such as CC3M and CC12M (Sharma et al., 2018; Changpinyo et al., 2021), generating forward and reverse edits along with corresponding edited captions (see Tab. 1, CCXM section).

## 5 EXPERIMENTS

## 5.1 EXPERIMENTAL SETUP

**Dataset Generation.** To train our method, we generate datasets consisting of paired forward and reverse instructions, as detailed in Sec. 4.2. For the initial experiments, we use the InstructPix2Pix dataset (Brooks et al., 2023), which provides generated image-caption pairs. We further extend our experiments to real-image datasets by fine-tuning GEMMA2 (Team et al., 2024). The real-image datasets include CC3M (Sharma et al., 2018) and CC12M (Changpinyo et al., 2021), for which we generate eight possible edits per image-caption pair. This increases diversity in the editing tasks, exposing the model to a wide range of transformations and enhancing its ability to generalize across different types of edits and real-world scenarios.

**Baselines.** We evaluate our method by comparing it against several baseline models. The primary baseline is InstructPix2Pix (IP2P) (Brooks et al., 2023), a supervised method that relies on ground-truth edited images during training. To demonstrate the advantages of our unsupervised approach, we train and test both IP2P and our model on the same datasets and compare their performance. We also compare our method with other instruction-based editing models, including MagicBrush (Zhang et al., 2023a), HIVE (Zhang et al., 2023b), MGIE (Fu et al., 2023), and SmartEdit (Huang et al., 2024). These additional comparisons allow us to evaluate how effectively our unsupervised model handles diverse and complex edits without the need for existing editing methods to generate ground-truth edited images or human-annotated data.

**Implementation Details.** Our method, UIP2P, fine-tunes SD-v1.5 model (Rombach et al., 2022), without any pre-training on supervised datasets. While we retain the IP2P architecture, our approach uses different training objectives, primarily focusing on enforcing Cycle Edit Consistency (CEC). Specifically, we employ the CLIP ViT-L/14 model, integrated into SD-v1.5, to calculate the losses. By using a single noise prediction across varying diffusion steps $t$ for forward and $\hat{t}$ for reverse, both sampled between 0-1000 (as proposed in IP2P training), our model reduces computational overhead, respect to IP2P (please refer to Sec. 5.4), while maintaining consistency between forward and reverse edits. This reuse of the prediction enables efficient and accurate editing with fewer inference steps than IP2P, which improves both generalization and performance, as empirically demonstrated in Sec. 5.4. UIP2P is trained using the AdamW optimizer (Loshchilov, 2017) with a batch size of 768 over 11K iterations. The base learning rate is set to 5e-05. All experiments are implemented in PyTorch (Paszke et al., 2019) and conducted on 16 NVIDIA H100 GPUs, with loss weights set as $\lambda_{\text{CLIP}} = 1.0$, $\lambda_{\text{attn}} = 0.5$, $\lambda_{\text{sim}} = 1.0$, and $\lambda_{\text{recon}} = 1.0$. We select the best configuration based on the validation loss of $\mathcal{L}_{\text{CEC}}$.

## 5.2 QUANTITATIVE RESULTS

### 5.2.1 IP2P TEST DATASET

We evaluate our method on the IP2P test split, containing 5K image-instruction pairs. Following Brooks et al. (2023), we use CLIP image similarity for visual fidelity and CLIP text-image similarity to assess alignment with the instructions. Higher scores in both metrics indicate better performance (upper right corner) by preserving image details (image similarity) and effectively applying the edits (direction similarity). As shown in the plot, UIP2P outperforms IP2P across both metrics. In these experiments, the text scale $s_T$ is fixed, while the image scale $s_I$ varies from 1.0 to 2.2.

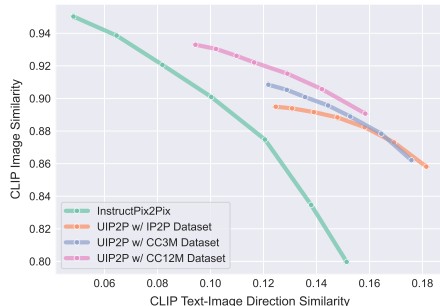

### 5.2.2 MAGICBRUSH TEST DATASET

The test split provides an evaluation pipeline with 535 sessions (source images for iterative editing) and 1053 turns (individual editing steps). It uses L1 and L2 norms for pixel accuracy, CLIP-I and DINO embeddings for image quality via cosine similarity, and CLIP-T to ensure alignment with local text descriptions.

Table 2: **Zero-shot Quantitative Comparison on MagicBrush (Zhang et al., 2023a) test set.** Instruction-based editing methods -are not fine-tuned on MagicBrush- are presented. In the multi-turn setting, target images are iteratively edited from the initial images. Best results are in **bold**.

| Settings | Methods | L1↓ | L2↓ | CLIP-I↑ | DINO↑ | CLIP-T↑ |
|---|---|---|---|---|---|---|
| **Single-turn** | HIVE (Zhang et al., 2023b) | 0.1092 | 0.0341 | 0.8519 | 0.7500 | 0.2752 |
| | InstructPix2Pix (Brooks et al., 2023) | 0.1122 | 0.0371 | 0.8524 | 0.7428 | 0.2764 |
| | UIP2P w/ IP2P Dataset | 0.0722 | 0.0193 | 0.9243 | 0.8876 | 0.2944 |
| | UIP2P w/ CC3M Dataset | 0.0680 | 0.0183 | 0.9262 | 0.8924 | **0.2966** |
| | UIP2P w/ CC12M Dataset | **0.0619** | **0.0174** | **0.9318** | **0.9039** | 0.2964 |
| **Multi-turn** | HIVE (Zhang et al., 2023b) | 0.1521 | 0.0557 | 0.8004 | 0.6463 | 0.2673 |
| | InstructPix2Pix (Brooks et al., 2023) | 0.1584 | 0.0598 | 0.7924 | 0.6177 | 0.2726 |
| | UIP2P w/ IP2P Dataset | 0.1104 | 0.0358 | 0.8779 | 0.8041 | 0.2892 |
| | UIP2P w/ CC3M Dataset | 0.1040 | 0.0337 | 0.8816 | 0.8130 | **0.2909** |
| | UIP2P w/ CC12M Dataset | **0.0976** | **0.0323** | **0.8857** | **0.8235** | 0.2901 |

As seen in Tab. 2, UIP2P perfoms the best for both single- and multi-turn settings. It is important to be noted that HIVE utilizes human feedback on edited images to understand user preferences and fine-tunes IP2P based on learned rewards, aligning the model more closely with human expectations. Table 2 also shows that increasing the number of samples in the training dataset and also training on real images provides better performance than training on the synthetic dataset, IP2P dataset.

### 5.2.3 USER STUDY

We conduct a user study on Prolific Platform (prolific) with 52 participants to evaluate six methods—IP2P, MagicBrush, HIVE, MGIE, SmartEdit, and UIP2P—across 15 image-edit instructions. For each instruction, participants select the best two methods, suggested in Huang et al. (2024), based on: (Q1)—how well the edit matched the instruction—and localization, and (Q2)—how accurately the edit was applied to the intended region. The table summarizes the percentage of times each method was chosen as a top performer for each question. UIP2P achieves the highest preference score, with MGIE and SmartEdit closely following. Unlike these methods, however, our approach introduces no latency penalty at inference time, offering both accuracy and efficiency.

Table 3: User Study.

| Models | (Q1) | (Q2) |
|---|---|---|
| IP2P | 8% | 12% |
| MagicBrush | 17% | 18% |
| HIVE | 14% | 13% |
| MGIE | 20% | 19% |
| SmartEdit | 19% | 18% |
| UIP2P | **22%** | **20%** |

### 5.3 QUALITATIVE RESULTS

We compare UIP2P with state-of-the-art methods, including InstructPix2Pix (Brooks et al., 2023), MagicBrush (Zhang et al., 2023a), HIVE (Zhang et al., 2023b), MGIE (Fu et al., 2023), and SmartEdit (Huang et al., 2024), on various datasets (Brooks et al., 2023; Zhang et al., 2023a; Shi et al., 2020; 2021). The tasks include color modifications, object removal, and structural changes. UIP2P consistently produces high-quality edits, applying transformations accurately while maintaining visual coherence. For example, in "let the bird turn yellow," UIP2P provides a more natural color change while preserving the bird's shape. Similar improvements are observed in tasks like "remove hot air balloons" and "change hat color to blue." These results demonstrate UIP2P's ability to handle diverse and complex edits, often matching or outperforming other methods.

### 5.4 ABLATION STUDY

**Loss Functions.** We conduct a zero-shot evaluation on the MagicBrush test set (single-turn setting) to assess the effectiveness of different loss functions. Starting with the base model which contains $\mathcal{L}_{CLIP}$ and $\mathcal{L}_{recon}$, we observe moderate performance across the same metrics. Adding $\mathcal{L}_{sim}$ allows the model to perform edits more freely, as the base configuration without it tends to create outputs similar to the input image.. Finally, $\mathcal{L}_{attn}$ enhances the model's focus on relevant regions and ensures that the region of interest remains consistent between the forward and reverse processes.

Table 4: Ablation on loss functions.

| Loss | L1↓ | L2↓ | CLIP-I↑ | DINO↑ | CLIP-T↑ |
|---|---|---|---|---|---|
| Base | 0.117 | 0.032 | 0.878 | 0.806 | **0.309** |
| + $\mathcal{L}_{sim}$ | 0.089 | 0.024 | 0.906 | 0.872 | 0.301 |
| + $\mathcal{L}_{attn}$ | **0.062** | **0.017** | **0.932** | **0.904** | 0.296 |

**Number of Steps During Inference.** We analyze the effect of varying the number of diffusion steps during inference. Fewer steps reduce computational time but may affect image quality. Our experiments show that UIP2P maintains high-quality edits with as few as 5 steps, providing a significant speedup without sacrificing accuracy. In contrast, IP2P requires more steps to achieve similar results. As shown in the figure, UIP2P consistently outperforms IP2P in both quality and efficiency, particularly with fewer inference steps.

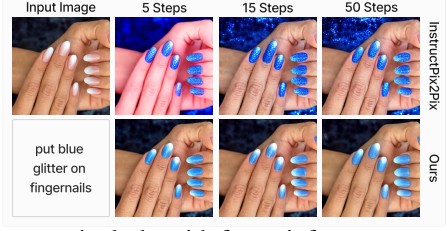

## 6 CONCLUSION

In this work, we present UIP2P, an unsupervised instruction-based image editing framework that leverages Cycle Edit Consistency (CEC) to ensure reversible and coherent edits without relying on ground-truth edited images. Some key components of our approach are Text and Image Direction Consistency, Attention Map Consistency, Reconstruction Consistency, and Unified Prediction with Varying Diffusion Steps, which together enforce consistency in both the image and attention space. Through experiments on real-image datasets, we show that UIP2P delivers high-quality and precise edits while maintaining the structure of the original image. It performs competitively against existing methods, demonstrating the effectiveness of our unsupervised approach, which scales efficiently across diverse editing tasks without the need for manually annotated datasets.

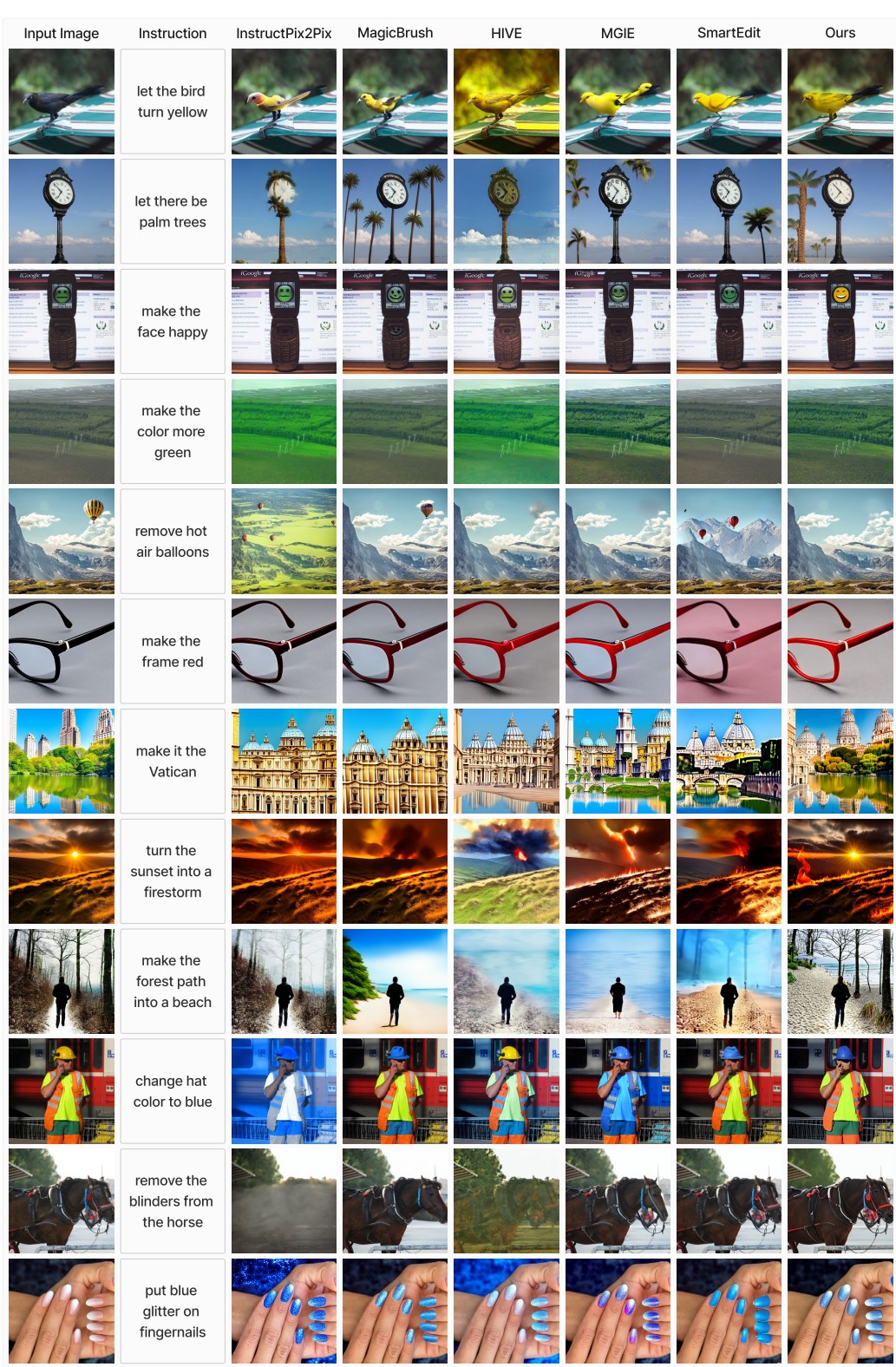

Figure 4: **Qualitative Examples.** UIP2P performance is shown across various tasks and datasets, compared to InstructPix2Pix, MagicBrush, HIVE, MGIE, and SmartEdit. Our method demonstrates either comparable or superior results in terms of accurately applying the requested edits while preserving visual consistency.

## 7 ETHICS STATEMENT

Advancements in localized image editing technology offer substantial opportunities to enhance creative expression and improve accessibility within digital media and virtual reality environments. Nonetheless, these developments also bring forth important ethical challenges, particularly concerning the misuse of such technology to create misleading content, such as deepfakes (Korshunov & Marcel, 2018), and its potential effect on employment in the image editing industry. Moreover, as also highlighted by Kenthapadi et al. (2023), it requires a thorough and careful discussion about their ethical use to avoid possible misuse. We believe that our method could help reduce some of the biases present in previous datasets, though it will still be affected by biases inherent in models such as CLIP. Ethical frameworks should prioritize encouraging responsible usage, developing clear guidelines to prevent misuse, and promoting fairness and transparency, particularly in sensitive contexts like journalism. Effectively addressing these concerns is crucial to amplifying the positive benefits of the technology while minimizing associated risks. In addition, our user study follows strict anonymity rules to protect the privacy of participants.

## 8 REPRODUCIBILITY STATEMENT

We aim to promote reproducibility by offering a clear description of our UIP2P method, including Cycle Edit Consistency (CEC). The complete algorithm can be found in Algorithm 1, along with pseudo-code to aid in replicating the implementation. In Appendix A.10, we explain the relevant frameworks and any modifications applied, ensuring compatibility with common tools. The reverse instruction datasets, will be made accessible along with the fine-tuned GEMMA2 model in a future release. Furthermore, Secs. 4.2 and 5.1 provide details on hyperparameters and the reverse instruction generation process. These sections outline the experimental setup and evaluation procedure to facilitate replication efforts.

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

# A  APPENDIX

TABLE OF CONTENTS

## A.1  RUNTIME ANALYSIS

Our method modifies the training objectives of IP2P by incorporating Cycle Edit Consistency (CEC) and additional loss functions. However, these changes do not affect the overall runtime. Inference time remains comparable to the original IP2P framework, as we retain the same architecture and model structure. Consequently, our approach introduces no additional complexity or overhead in terms of processing time or resource consumption. This gives UIP2P an advantage over methods like MGIE (Fu et al., 2023) and SmartEdit Huang et al. (2024), which rely on large language models (LLMs) during inference in terms of runtime and resource consumption.

Additionally, as shown in Sec. 5.4, UIP2P requires fewer inference steps to achieve accurate edits. For instance, while IP2P typically uses more steps, *e.g.*, from 50 to 100 steps, UIP2P can produce coherent results in as few as five steps. This reduction in steps leads to faster inference times, offering a clear efficiency advantage without compromising on quality, especially in real-time or large-scale applications.

## A.2  ABLATION STUDY ON LOSS FUNCTIONS

We focused our ablation studies on $\mathcal{L}_{\text{sim}}$ and $\mathcal{L}_{\text{attn}}$ because these losses are additional components beyond the core $\mathcal{L}_{\text{CLIP}}$ and $\mathcal{L}_{\text{recon}}$. The core losses are essential for ensuring semantic alignment and reversibility in Cycle Edit Consistency (CEC), forming the foundation of our method. Without $\mathcal{L}_{\text{CLIP}}$ and $\mathcal{L}_{\text{recon}}$, the model risks diverging, losing its ability to preserve both the input's structure and its semantic coherence during edits.

Adding $\mathcal{L}_{\text{sim}}$ enables the model to perform edits more freely by encouraging alignment between image and textual embeddings, thereby expanding its capacity for complex and diverse transformations. On the other hand, $\mathcal{L}_{\text{attn}}$ refines the model's ability to focus on relevant regions during edits, improving localization and reducing unintended changes in non-targeted areas.

$\mathcal{L}_{\text{CLIP}}$ is applied between the input image and the edited image to ensure semantic alignment with the edit instruction. The reconstructed image is already constrained by $\mathcal{L}_{\text{recon}}$, which enforces structural and semantic consistency with the input. Adding $\mathcal{L}_{\text{CLIP}}$ to the reconstructed image would be redundant and could interfere with the reversibility objective. Our design does not apply $\mathcal{L}_{\text{CLIP}}$ to the reconstructed image to preserve the focus on reversibility and prevent conflicting optimization objectives.

### A.3 DISCUSSION ON REDUCED DDIM STEPS

This observation is based on empirical results, as detailed in **Number of Steps During Inference** (Sec. 5.4). Specifically, we hypothesize that the CEC ensures strong alignment between forward and reverse edits, enabling the model to produce high-quality outputs even with fewer DDIM steps. Additionally, as shown in Algorithm 1 (Lines 4 and 8), our method uses the same denoising prediction across all timesteps to recover the image, which enhances efficiency.

In contrast, IP2P does not optimize its losses in image space during training, limiting its ability to achieve comparable results with fewer DDIM steps. This reduction in DDIM steps contributes to improved scalability and makes our method more applicable in real-world scenarios where computational resources are often constrained.

### A.4 DETAILS OF COMPETITOR METHODS

Our method offers significant advantages over competitors in both training and inference. Unlike supervised methods that rely on paired triplets of input images, edited images, and instructions, our approach eliminates the need for such datasets, reducing biases and improving scalability. For example, MagicBrush is fine-tuned on a human-annotated dataset, while HIVE leverages Prompt-to-Prompt editing with human annotators, introducing dependency on labor-intensive processes. Furthermore, MGIE and SmartEdit rely on LLMs during inference, which significantly increases computational overhead. These distinctions highlight the efficiency and practicality of our approach, as it avoids the need for expensive human annotations and additional inference-time complexities. Like other editing methods, our approach can produce small variations for different random seeds but consistently applies the specified edit, eliminating the need for manual selection. To the best of our knowledge, the compared methods (*e.g.*, MagicBrush, InstructPix2Pix) also do not involve manual selection.

**InstructPix2Pix** (Brooks et al., 2023)[1]: InstructPix2Pix (IP2P) is a diffusion-based model that performs instruction-based image editing by training on triplets of input image, instruction, and edited image. The model is fine-tuned on a synthetic dataset of edited images generated by combining large language models (LLMs) and Prompt-to-Prompt (Hertz et al., 2022). This approach relies on paired datasets, which can introduce biases and limit generalization. InstructPix2Pix serves as one of the key baselines for our comparison, given its supervised training methodology.

**HIVE** (Zhang et al., 2023b)[2]: HIVE is another instruction-based editing model that fine-tunes InstructPix2Pix based on human feedback. Specifically, HIVE learns from user preferences about which edited images are preferred, incorporating this feedback into the model training. While this approach allows HIVE to better align with human expectations, it still builds on top of InstructPix2Pix and does not start training from scratch. This limits its flexibility compared to methods like UIP2P, which are trained from the ground up.

**MagicBrush** (Zhang et al., 2023a)[3]: MagicBrush fine-tunes the pre-trained weights of InstructPix2Pix on a human-annotated dataset to improve real-image editing performance. While this fine-tuning approach makes MagicBrush highly effective for specific tasks with ground-truth labels, it limits its generalizability compared to methods like UIP2P, which are trained from scratch. Moreover, MagicBrush's reliance on human-annotated data introduces significant scalability challenges, as obtaining such annotations is both costly and labor-intensive. This dependency makes it less suited for broader datasets where large-scale annotations may not be feasible.

**MGIE** (Fu et al., 2023)[4]: MGIE introduces a large multimodal language model to generate more precise instructions for image editing. Like InstructPix2Pix, MGIE requires a paired dataset for training but uses the language model to improve the quality of the instructions during inference. However, this reliance on LLMs during inference adds computational overhead. In contrast, UIP2P operates without LLMs at inference time, reducing overhead while maintaining flexibility.

---

[1]https://github.com/timothybrooks/instruct-pix2pix
[2]https://github.com/salesforce/HIVE
[3]https://github.com/OSU-NLP-Group/MagicBrush
[4]https://ml-mgie.com/playground.html

**SmartEdit** (Huang et al., 2024)[5]: SmartEdit is based on InstructDiffusion, a model already trained for instruction-based image editing tasks. It introduces a bidirectional interaction module to improve text-image alignment, but its reliance on the pre-trained InstructDiffusion limits flexibility, as SmartEdit does not start training from scratch. Additionally, SmartEdit depends on large language models (LLMs) during inference, increasing computational overhead. This makes SmartEdit less efficient than UIP2P in scenarios where real-time or large-scale processing is required.

During evaluation, we use the publicly available implementations and demo pages of the baseline methods. Each baseline provides a different approach to instruction-based image editing, and together they offer a comprehensive set of methods for comparing the performance, flexibility, and efficiency of the proposed method, UIP2P.

## A.5 Cycle Edit Consistency Example

We demonstrate CEC with a visual example during inference. In the forward pass, the model transforms the input image based on the instruction (*e.g.*, "turn the forest path into a beach"). In the reverse pass, the corresponding reverse instruction (*e.g.*, "turn the beach back into a forest") is applied, reconstructing the original image. This showcases the model's ability to maintain consistency and accuracy across complex edits, ensuring that both the forward and reverse transformations align coherently. Additional examples, such as adding and removing objects, further emphasize UIP2P's adaptability in diverse editing tasks. This example illustrates how our method ensures precise, reversible edits while maintaining the integrity of the original content.

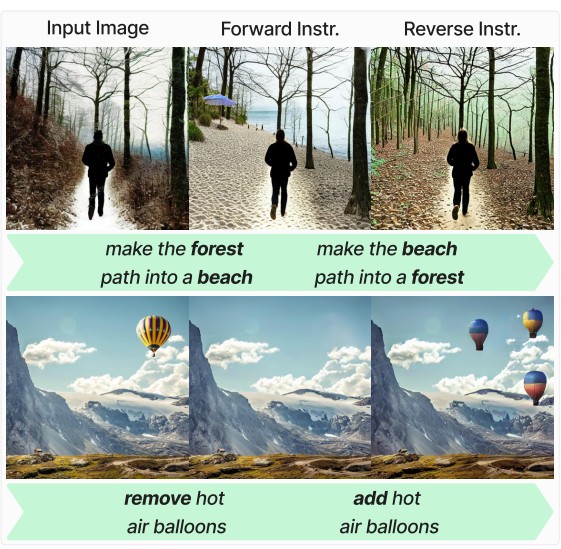

## A.6 Evaluation on the PIE Benchmark

We apply our method to the PIE benchmark to evaluate its performance on diverse editing tasks and compare it to IP2P, a representative feed-forward instruction-based editing method and a supervised alternative to our approach. The table below summarizes the results:

| Methods | Distance ↓ | PSNR ↑ | LPIPS ↓ | MSE ↓ | SSIM ↑ | Whole ↑ | Edit ↑ |
|---------|-----------|--------|---------|-------|--------|---------|--------|
| IP2P | 57.91 | 20.82 | 158.63 | 227.78 | 76.26 | 23.61 | 21.64 |
| Ours | **27.05** | **26.85** | **60.57** | **40.07** | **83.69** | **24.78** | **21.89** |

Table 5: **Performance comparison on the PIE benchmark.** Lower values for Distance, LPIPS, and MSE indicate better performance, while higher values for PSNR, SSIM, Whole, and Edit indicate improved quality and structural preservation.

The results show that our method outperforms IP2P across most metrics, including better preservation of structure (PSNR and SSIM), lower perceptual differences (LPIPS), and reduced mean squared error (MSE). These improvements demonstrate the scalability and versatility of our approach on a broader benchmark. This analysis is included in the revised manuscript to provide a comprehensive evaluation of our method.

---

[5]https://github.com/TencentARC/SmartEdit

### A.7 Attention Consistency Across Noise Steps in Training

At training time, we sample two different noise steps for the forward and backward processes, which are conditioned on the input image and edit instruction. Attention consistency is enforced between these different noise steps to ensure that the model attends to the same regions during both forward and reverse edits. This is supported by the observation that cross-attention scores in instruction-based editing methods tend to be more consistent across timesteps, as the edit instruction remains fixed and the model's focus shifts only to the regions being edited (see Fig. 5).

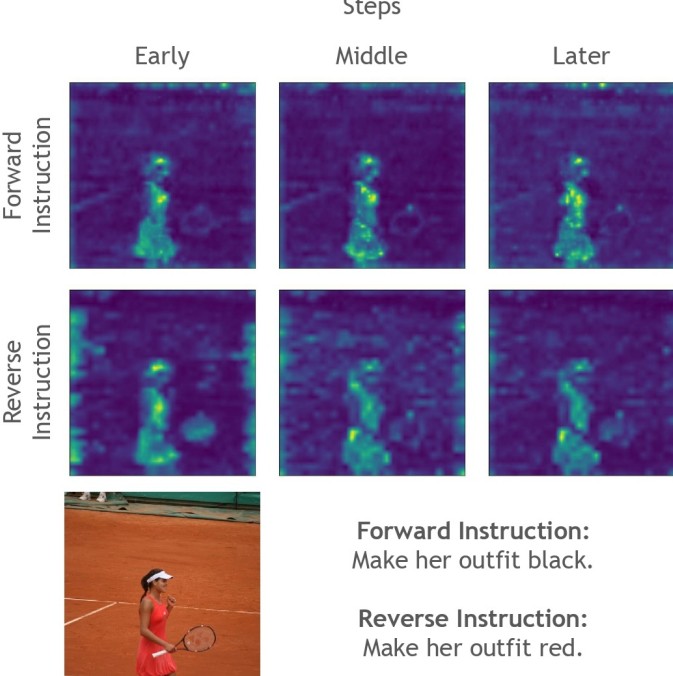

Figure 5: **Attention maps for diffusion steps.** Cross-attention maps for forward (top) and reverse (middle) instructions across early, middle, and later noise steps. The model enforces attention consistency, focusing on relevant regions for both edits.

Recent works, such as those by Guo et al. Guo & Lin (2024) and Simsar et al. Simsar et al. (2023), demonstrate that regularizing attention space with a mask during inference enables localized edits in IP2P. Our method builds on these ideas by incorporating attention consistency into the training phase, making it possible to focus on relevant regions from the start and avoiding the need for additional inference-time modifications.

### A.8 Additional Quantitative Analysis on MagicBrush Test

In this section, we present the full quantitative analysis on the MagicBrush test set, including results from both global description-guided and instruction-guided models, as shown in Tab. 6. While our method, UIP2P, is not fine-tuned on human-annotated datasets like MagicBrush, it still achieves highly competitive results compared to models specifically fine-tuned for the task. In particular, UIP2P demonstrates either the best or second-best performance in key metrics such as L1, L2, and CLIP-I, even outperforming fine-tuned models in several cases. This highlights the robustness and generalization capabilities of UIP2P, showing that it can effectively handle complex edits without the need for specialized training on real datasets. These results further validate that UIP2P delivers high-quality edits in a variety of contexts, maintaining competitive performance against fine-tuned models on the MagicBrush dataset, which is human-annotated.

Table 6: **Quantitative comparison on MagicBrush (Zhang et al., 2023a) test set.** In the multi-turn setting, target images are iteratively edited from the initial source images. Best results are in **bold**.

| Settings | Methods | L1↓ | L2↓ | CLIP-I↑ | DINO↑ | CLIP-T↑ |
|---|---|---|---|---|---|---|
| | *Global Description-guided* | | | | | |
| | Open-Edit (Liu et al., 2020) | 0.1430 | 0.0431 | 0.8381 | 0.7632 | 0.2610 |
| | VQGAN-CLIP (Crowson et al., 2022) | 0.2200 | 0.0833 | 0.6751 | 0.4946 | **0.3879** |
| | SD-SDEdit (Meng et al., 2022) | 0.1014 | 0.0278 | 0.8526 | 0.7726 | 0.2777 |
| | Text2LIVE (Bar-Tal et al., 2022) | 0.0636 | **0.0169** | 0.9244 | 0.8807 | 0.2424 |
| | Null Text Inversion (Mokady et al., 2022) | 0.0749 | 0.0197 | 0.8827 | 0.8206 | 0.2737 |
| Single-turn | *Instruction-guided* | | | | | |
| | HIVE (Zhang et al., 2023b) | 0.1092 | 0.0341 | 0.8519 | 0.7500 | 0.2752 |
| | w/ MagicBrush (Zhang et al., 2023a) | 0.0658 | 0.0224 | 0.9189 | 0.8655 | 0.2812 |
| | InstructPix2Pix (Brooks et al., 2023) | 0.1122 | 0.0371 | 0.8524 | 0.7428 | 0.2764 |
| | w/ MagicBrush (Zhang et al., 2023a) | 0.0625 | 0.0203 | **0.9332** | 0.8987 | 0.2781 |
| | UIP2P w/ IP2P Dataset | 0.0722 | 0.0193 | 0.9243 | 0.8876 | 0.2944 |
| | UIP2P w/ CC3M Dataset | 0.0680 | 0.0183 | 0.9262 | 0.8924 | 0.2966 |
| | UIP2P w/ CC12M Dataset | **0.0619** | 0.0174 | 0.9318 | **0.9039** | 0.2964 |
| | *Global Description-guided* | | | | | |
| | Open-Edit (Liu et al., 2020) | 0.1655 | 0.0550 | 0.8038 | 0.6835 | 0.2527 |
| | VQGAN-CLIP (Crowson et al., 2022) | 0.2471 | 0.1025 | 0.6606 | 0.4592 | **0.3845** |
| | SD-SDEdit (Meng et al., 2022) | 0.1616 | 0.0602 | 0.7933 | 0.6212 | 0.2694 |
| | Text2LIVE (Bar-Tal et al., 2022) | 0.0989 | **0.0284** | 0.8795 | 0.7926 | 0.2716 |
| | Null Text Inversion (Mokady et al., 2022) | 0.1057 | 0.0335 | 0.8468 | 0.7529 | 0.2710 |
| Multi-turn | *Instruction-guided* | | | | | |
| | HIVE (Zhang et al., 2023b) | 0.1521 | 0.0557 | 0.8004 | 0.6463 | 0.2673 |
| | w/ MagicBrush (Zhang et al., 2023a) | 0.0966 | 0.0365 | 0.8785 | 0.7891 | 0.2796 |
| | InstructPix2Pix (Brooks et al., 2023) | 0.1584 | 0.0598 | 0.7924 | 0.6177 | 0.2726 |
| | w/ MagicBrush (Zhang et al., 2023a) | **0.0964** | 0.0353 | **0.8924** | **0.8273** | 0.2754 |
| | UIP2P w/ IP2P Dataset | 0.1104 | 0.0358 | 0.8779 | 0.8041 | 0.2892 |
| | UIP2P w/ CC3M Dataset | 0.1040 | 0.0337 | 0.8816 | 0.8130 | 0.2909 |
| | UIP2P w/ CC12M Dataset | 0.0976 | 0.0323 | 0.8857 | 0.8235 | 0.2901 |

## A.9 USER STUDY SETTING

We conduct a user study with 52 anonymous participants on the Prolific Platform (prolific), presenting them with 30 questions. Each question shows participants six edited images generated by different methods, alongside their corresponding input images and edit instructions. Participants are tasked with evaluating the effectiveness of the edits in achieving the specified outcome (Q1) and assessing the ability of the editing method to preserve the details in areas not targeted by the instruction (Q2).

For example, as shown in Fig. 6, where the edit instruction is *make the face happy*, participants are asked to determine which of the six edited images (a-f) best satisfies the instruction while maintaining the fidelity of irrelevant details in the scene. By aggregating responses from participants, we gather insights into the preferred methods for both accurate editing and detail preservation. This feedback provides a fair comparison between methods, complementing the quantitative analysis.

## A.10 ADDITIONAL IMPLEMENTATION DETAILS

### A.10.1 CODE IMPLEMENTATION OVERVIEW

Our UIP2P implementation with CEC builds on existing frameworks for reproducibility:

- **Base Framework:** The code is based on InstructPix2Pix[6], which provides the foundation for instruction-based image editing.

---

[6] https://github.com/timothybrooks/instruct-pix2pix

Figure 6: **User Study Setup.** The input image is shown alongside randomly ordered edited images generated by different methods (a)-(f) based on the edit instruction, "make the face happy." Participants are asked to select the best two methods that match the editing effect and those that best preserve irrelevant instruction regions.

- **Adopted CLIP Losses:** We adopted and modified CLIP-based loss functions from StyleGAN-NADA[7] to fit CEC, improving image-text alignment for our specific tasks.

### A.10.2 ALGORITHM OVERVIEW

In this section, we explain the proposed method, UIP2P, which introduces unsupervised learning for instruction-based image editing. The core of our approach is the Cycle Edit Consistency (CEC), which ensures that edits are coherent and reversible when cycled through both forward and reverse instructions.

The algorithm consists of two key processes:

- **Forward Process:** Starting with an input image and a forward edit instruction, noise is first added to the image. The model then predicts the noise, which is applied to reverse the noise process and recover the edited image (*see Algorithm 1, lines 2-4*).
- **Backward Process:** Given the forward-edited image and a reverse edit instruction, noise is applied again. The model predicts the reverse noise, which is used to undo the edits and reconstruct the original image. This ensures that the reverse edits are consistent with the original input image (*see Algorithm 1, lines 6-8*).

CEC is applied between the original input image, the forward-edited image, and the reconstructed image, along with their respective attention maps and captions (*see Algorithm 1, line 10*). The $\mathcal{L}_{CEC}$ function guides the model's learning through backpropagation (*see Algorithm 1, lines 12-13*).

The complete algorithm for the UIP2P method is outlined in **Algorithm 1**.

---

**Algorithm 1** Unsupervised Instruction-Based Image Editing (UIP2P) with CEC

---

**Require:** Image $I_{input}$ (input image), Forward edit instruction $F$, Reverse edit instruction $R$, Noise levels $t$ (forward), $\hat{t}$ (backward), Model $M$, Loss function $L_{CEC}$, Noise function $N$, Input caption $T_{input}$, Edited caption $T_{edit}$
**Ensure:** Edited image $I_{edit}$, Reconstructed image $I_{recon}$

1: **Forward Process:**
2: $z_t \leftarrow N(I_{input}, t)$              ▷ Add noise $t$ to the input image $I_{input}$
3: $\hat{\epsilon}_F, A_f \leftarrow M(z_t | I_{input}, F)$ ▷ Model $M$ predicts forward noise $\hat{\epsilon}_F$ and extracts attention map $A_f$
4: $I_{edit} \leftarrow \text{Apply}(\hat{\epsilon}_F, z_t, t)$ ▷ Apply predicted noise $\hat{\epsilon}_F$ to reverse the process of obtaining $z_t$ and recover $I_{edit}$

5: **Backward Process:**
6: $z_{\hat{t}} \leftarrow N(I_{edit}, \hat{t})$             ▷ Add noise $\hat{t}$ to the forward-edited image $I_{edit}$
7: $\hat{\epsilon}_R, A_r \leftarrow M(z_{\hat{t}} | I_{edit}, R)$ ▷ Model $M$ predicts reverse noise $\hat{\epsilon}_R$ and extracts attention map $A_r$
8: $I_{recon} \leftarrow \text{Apply}(\hat{\epsilon}_R, z_{\hat{t}}, \hat{t})$ ▷ Apply predicted noise $\hat{\epsilon}_R$ to reverse the process of obtaining $z_{\hat{t}}$ and recover $I_{recon}$

9: **Cycle Edit Consistency Loss:**
10: $L_{CEC} \leftarrow L(I_{input}, I_{edit}, I_{recon}, A_f, A_r, T_{input}, T_{edit})$     ▷ Compute CEC loss using $I_{input}$, $I_{edit}$, $I_{recon}$, attention maps $A_f$, $A_r$, input text $T_{input}$, and edited text $T_{edit}$

11: **Update Model:**
12: Backpropagate the loss $L_{CEC}$ and update the model $M$
13: Repeat until convergence

---

### A.11 DATASET FILTERING

We apply CLIP (Radford et al., 2021a) to both the CC3M (Sharma et al., 2018) and CC12M (Changpinyo et al., 2021) datasets to calculate the similarity between captions and images, ensuring that

---

[7]https://github.com/rinongal/StyleGAN-nada

the text descriptions accurately reflect the content of the corresponding images. Following the methodology used in InstructPix2Pix (IP2P) (Brooks et al., 2023), we adopt a CLIP-based filtering strategy with a similarity threshold set at 0.2. This threshold filters out image-caption pairs that do not have sufficient semantic alignment, allowing us to curate a dataset with higher-quality text-image pairs. For the filtering process, we utilize the CLIP ViT-L/14 model, which provides a robust and well-established framework for capturing semantic similarity across text and images.

By applying this filtering process, we ensure that only relevant and coherent pairs remain in the dataset, improving the quality of training data and helping the model better generalize to real-world editing tasks. As a result, the filtered CC3M dataset contains 2.5 million image-caption pairs, while the filtered CC12M dataset contains 8.5 million pairs. This careful curation of the dataset enhances the reliability of the training process without relying on human annotations, making it scalable for broader real-image datasets without the cost and limitations of human-annotated ground-truth datasets (Brooks et al., 2023; Zhang et al., 2023a).

### A.12   MORE EXAMPLES FROM REVERSE INSTRUCTIONS DATASET

To demonstrate the versatility of our reverse instruction dataset, we provide examples with multiple variations of edits for two different input captions. Each caption has four distinct edits, such as color changes, object additions, object removals, and positional adjustments. This variety helps the model generalize across a wide range of tasks and scenarios, as discussed in Sec. 4.2. The use of LLMs to generate reverse instructions further enhances the flexibility of our dataset.

Table 7: **Examples of Four Possible Edits for Two Different Input Captions.** Our dataset generation process showcases the flexibility of the reverse instruction dataset by demonstrating multiple transformations for the same caption.

| Input Caption | Edit Instruction | Edited Caption | Reverse Instruction |
|---|---|---|---|
| A dog sitting on a couch | change the dog's color to brown | A brown dog sitting on a couch | change the dog's color back to white |
|  | add a ball next to the dog | A dog sitting on a couch with a ball | remove the ball |
|  | remove the dog | An empty couch | add the dog back |
|  | move the dog to the floor | A dog sitting on the floor | move the dog back to the couch |
| A car parked on the street | change the car color to red | A red car parked on the street | change the car color back to black |
|  | add a bicycle next to the car | A car parked on the street with a bicycle | remove the bicycle |
|  | remove the car | An empty street | add the car back |
|  | move the car to the garage | A car parked in the garage | move the car back to the street |

These examples, along with others in Tab. 1, illustrate the diversity of edit types our model learns, enabling it to perform a wide range of tasks across different real-image datasets. The reverse instruction mechanism ensures that the edits are reversible, maintaining consistency and coherence in both the forward and reverse transformations.

