# OpenReview forum: "UIP2P: Unsupervised Instruction-Based Image Editing via Cycle Edit Consistency"
_ICLR.cc/2025/Conference — Submitted to ICLR 2025_

### Official Review · Reviewer_goKN · 2024-10-24

**Soundness:** 3
**Presentation:** 3
**Contribution:** 3
**Rating:** 6
**Confidence:** 4

**Summary:**

This method presents a new method for instruct-based image editing. Different from the previous method, this method does not require paired editing images for training via cycle edit consistency. By training on the real images only, the overall results show much better performance than the previous supervised methods.

**Strengths:**

- Editing the images via instruction is natural. The dataset collection is expensive. This method provides an unsupervised method to achieve this goal and several loss functions are utilized to make it happen.
- The experiment results are detailed. Different datasets are trained to show the differences. Single-turn and multi-turn editing are considered.

**Weaknesses:**

- This method involves several loss functions for training, which might make it hard to find the best checkpoint.
- There are many loss functions in UIP2P. However, the author only does the ablations on the L_sim and  L_attn. What about other loss functions?

**Questions:**

1. Are there multi-turn results contained in the paper?
2. UIP2P is trained based on instruct-pix2pix. Can UIP2P be trained from scratch (e.g. Stable Diffusion checkpoint)? Or Does this method require paired instructions pre-trained?
3. It is interesting to see that UIP2P can edit the image in fewer DDIM steps. Why? Any possible discussions?
4. In Fig.3. I am confused about clip Loss. The clip loss is added between the input caption, edited caption, input image, and edited image. Why this loss is added only between the input image and the edited image? What about the clip loss on the reconstructed image?

---

> ### Author Response · Authors · 2024-11-20
>
> ## Weaknesses
>
> > This method involves several loss functions for training, which might make it hard to find the best checkpoint.
>
> We monitor CLIP similarity (semantic alignment) and reconstruction loss (structural preservation) on a validation set to select the checkpoint with the best trade-off between semantic alignment and structural preservation by measuring the validation losses. This is similar to best practices followed by all ML models to monitor convergence. We also show the contributions of each loss function through an ablation study in Tab. 4. This process ensures robust performance, and we included it in the revised manuscript, see Lines 389-391.
>
> > There are many loss functions in UIP2P. However, the author only does the ablations on the L_sim and L_attn. What about other loss functions?
>
> We focused our ablation studies on $ \mathcal{L}\_{\text{sim}} $ and $ \mathcal{L}\_{\text{attn}} $ because these losses are additional components beyond the core $ \mathcal{L}\_{\text{CLIP}} $ and $ \mathcal{L}\_{\text{recon}} $. The core losses are critical for ensuring semantic alignment and reversibility in Cycle Edit Consistency (CEC), forming the foundation of our method; without them, the model risks diverging. Adding $ \mathcal{L}\_{\text{sim}} $ allows the model to perform edits more freely, while $ \mathcal{L}\_{\text{attn}} $ enhances focus on relevant regions, improving localization. To clarify further, we updated the manuscript; see Sec. 5.4 and Sec A.2.
>
> ## Questions
> > 1. Are there multi-turn results contained in the paper?
>
> Yes, multi-turn editing is evaluated in Tab. 2, demonstrating the method’s robustness with sequential instructions. Additionally, Sec. A.5 shows forward-reverse concatenations, effectively representing multi-turn editing. We clarified this connection in the revised manuscript to better highlight the multi-turn capabilities of our method.
>
> > 2. UIP2P is trained based on instruct-pix2pix. Can UIP2P be trained from scratch (e.g. Stable Diffusion checkpoint)? Or Does this method require paired instructions pre-trained?
>
> UIP2P is not pre-trained on any datasets that contain ground-truth edited images. Our method starts from the Stable Diffusion v1.5 checkpoint, as stated in Lines 378–379 of the manuscript. This ensures that the model does not rely on pre-trained supervision with triplet datasets of input, edited images, and instructions, differentiating it from methods like InstructPix2Pix. We updated the manuscript to avoid any confusion, Lines 379–380.
>
> > 3. It is interesting to see that UIP2P can edit the image in fewer DDIM steps. Why? Any possible discussions?
>
> This observation was empirical, and we provide an intuition for this behavior in Lines 265–270 of the manuscript. Specifically, we hypothesize that the Cycle Edit Consistency (CEC) framework ensures strong alignment between forward and reverse edits, which enables the model to produce high-quality outputs even with fewer DDIM steps. Additionally, as shown in Algorithm 1 (Lines 4 and 8), our method uses the same denoising prediction across all timesteps to recover the image, which enhances efficiency. In contrast, IP2P does not optimize its losses in image space during training. This reduction in DDIM steps contributes to improved scalability and makes our method more applicable in real-world scenarios where computational resources are often limited. To further clarify this, we updated the manuscript, see Sec. A.3, which emphasizes the efficiency and practical advantages of our approach.
>
> > 4. In Fig.3. I am confused about clip Loss. The clip loss is added between the input caption, edited caption, input image, and edited image. Why this loss is added only between the input image and the edited image? What about the clip loss on the reconstructed image?
>
> The CLIP loss is applied between the input image and the edited image to ensure semantic alignment with the edit instruction. The reconstructed image is already constrained by $ \mathcal{L}\_{\text{recon}} $, which enforces structural and semantic consistency with the input. Adding a CLIP loss to the reconstructed image would be redundant and could interfere with the reversibility objective. To further clarify this, we updated the manuscript, see Sec. A.2.

---

> > ### Author Response · Authors · 2024-11-29
> >
> > Dear Reviewer goKN, we are thankful for your review. As the rebuttal process is coming to an end, please let us know if your concerns are well addressed. We are happy to provide further clarification.

---

> ### Author Response · Authors · 2024-12-02
>
> Dear Reviewer goKN,
>
> We are now in the last 24 hours of the rebuttal period for reviewers. We have addressed all your concerns in our response and revised the manuscript accordingly. If you have any remaining questions or need further clarification, please do not hesitate to reach out.
>
> Thank you again for your valuable feedback and time.
>
> Kind regards,
>
> Paper 1784 Authors

---

> > ### Author Response · Authors · 2024-12-03
> >
> > Dear Reviewer goKN, we are thankful for your review. As the rebuttal deadline is coming to an end, please let us know if your concerns are well addressed. We are happy to provide further clarification.

---

### Official Review · Reviewer_MJQy · 2024-11-01

**Soundness:** 3
**Presentation:** 3
**Contribution:** 3
**Rating:** 6
**Confidence:** 4

**Summary:**

This paper proposes an unsupervised model for instruction-based image editing. Existing supervised methods rely on datasets consisting of input images, edited images, and editing instructions, which limits their generalization. This paper introduces Cycle Edit Consistency (CEC) to avoid this heavy dependency. Specifically, the authors apply a forward and backward edit in one training step and enforce consistency in both the image and attention space with CLIP.

**Strengths:**

1. This paper is well written and easy to follow.
2. The task is interesting and the problem setting is reasonable.
3. Although the idea of cycle consistency has been seen in CycleGAN, proving that it can be successfully applied to instruction-based image editing also represents a certain contribution.

**Weaknesses:**

Due to the lack of GroundTruth, it is difficult to make an accurate comparison of the experimental results. Looking at the edited image results, the proposed method does not seem to have a clearly superior advantage compared to the methods against which it was benchmarked. For example, in the first row of Figure 4, I prefer the result of HIVE; in the second row, I favor MagicBrush; in the third row, I like the results of MagicBrush and MGIE; in the sixth row, I am more inclined towards MGIE's result; in the eighth row, I prefer MAIE's outcome; and in the last row, I favor the results of InstructPix2Pix and SmartEdit.

**Questions:**

1. One thing I am curious about is whether the proposed method generates different editing results for the same input during inference. Is there a need to manually select the better images? Similarly, do the compared methods also select one result from multiple outcomes for comparison?
2. I noticed that the backbone structure of the proposed method is InstructPix2Pix. Therefore, during training, does this method use the pre-trained weights of InstructPix2Pix, or does it start training from scratch?

---

> ### Author Response · Authors · 2024-11-20
>
> > Due to the lack of GroundTruth, it is difficult to make an accurate comparison of the experimental results. Looking at the edited image results, the proposed method does not seem to have a clearly superior advantage compared to the methods against which it was benchmarked. For example, in the first row of Figure 4, I prefer the result of HIVE; in the second row, I favor MagicBrush; in the third row, I like the results of MagicBrush and MGIE; in the sixth row, I am more inclined towards MGIE's result; in the eighth row, I prefer MAIE's outcome; and in the last row, I favor the results of InstructPix2Pix and SmartEdit.
>
> We acknowledge the subjective nature of qualitative evaluation in instruction-based image editing, as preferences can vary depending on individual perception. To address this, we conducted a user study to aggregate diverse opinions, providing a more comprehensive comparison across six methods. Our method achieved the highest performance, with 22% for Q1 and 20% for Q2. Combined with strong quantitative metrics, all of which are calculated using ground-truth (GT) edited images, these evaluations demonstrate our method's competitive performance and robustness.
>
> Additionally, our method offers significant advantages over competitors in both training and inference. Unlike supervised methods that rely on paired triplets of input images, edited images, and instructions, our approach eliminates the need for such datasets, reducing biases and improving scalability. For example, MagicBrush is fine-tuned on a human-annotated dataset, while HIVE uses Prompt-to-Prompt with human annotators. Furthermore, MGIE and SmartEdit rely on large language models (LLMs) during inference, significantly increasing computational overhead. We expanded on these points in the revised manuscript, see Sec. A.4.
>
> ## Questions
>
> > 1. One thing I am curious about is whether the proposed method generates different editing results for the same input during inference. Is there a need to manually select the better images? Similarly, do the compared methods also select one result from multiple outcomes for comparison?
>
> Like other editing methods, our approach can produce small variations for different random seeds but consistently applies the specified edit; therefore, there is no need for manual selection. To the best of our knowledge, the compared methods (e.g., MagicBrush, InstructPix2Pix) also do not involve manual selection. We clarified this in the revised manuscript, see Sec. A.4.
>
> > 2. I noticed that the backbone structure of the proposed method is InstructPix2Pix. Therefore, during training, does this method use the pre-trained weights of InstructPix2Pix, or does it start training from scratch?
>
> No, our method does not use IP2P pre-trained weights. Our method starts from the Stable Diffusion v1.5 checkpoint, as stated in Lines 378–380 of the manuscript. It does not require the pre-trained weights of InstructPix2Pix.

---

> > ### Author Response · Authors · 2024-11-29
> >
> > Dear Reviewer MJQy, we are thankful for your review. As the rebuttal process is coming to an end, please let us know if your concerns are well addressed. We are happy to provide further clarification.

---

> ### Author Response · Authors · 2024-12-02
>
> Dear Reviewer MJQy,
>
> We are now in the last 24 hours of the rebuttal period for reviewers. We have addressed all your concerns in our response and revised the manuscript accordingly. If you have any remaining questions or need further clarification, please do not hesitate to reach out.
>
> Thank you again for your valuable feedback and time.
>
> Kind regards,
>
> Paper 1784 Authors

---

> > ### Author Response · Authors · 2024-12-03
> >
> > Dear Reviewer MJQy, we are thankful for your review. As the rebuttal deadline is coming to an end, please let us know if your concerns are well addressed. We are happy to provide further clarification.

---

### Official Review · Reviewer_aTp8 · 2024-11-02

**Soundness:** 2
**Presentation:** 2
**Contribution:** 3
**Rating:** 5
**Confidence:** 5

**Summary:**

The paper is motivated to remove the dependency on the triplet data of before-and-after images and editing instructions in training an image editing model and also aims to remove the bias introduced by the automatically created triplet data, which is meaningful and objective. To achieve this, the author proposes the cycle edit consistency training that does not require the edited image. Specifically, the author designs several consistency losses in various aspects such as images, pixels, and attention. The author evaluates the method on magic brush test data.

**Strengths:**

1. The motivation for removing the triplet is very meaningful and reasonable in instruction-based image editing. The bias of training data is also an important problem.

2. The presentation is easy to understand and the discussion for the motivation is reasonable and convincing.

3. There are several experiment evaluation types including quantitative, qualitative, and user study.

**Weaknesses:**

Weaknesses and Questions:

1.	Backpropagation efficiency: The author proposes several consistency losses. Considering the diffusion process InsP2P includes multiple steps to get the final image, what is the backpropagation process of the loss constructed on the image? I think this will be a recursive procedure since the loss will have to backpropagate from the last step of diffusion to the very first step which is insanely computationally expensive and requires large GPU memory.  The same questions apply to attention map consistency, etc.

2.	Validation for removing bias in InsP2P apart from inconsistency. I think the proposed method can maintain the consistency of forward and reversed edits but cannot guarantee that the edited results are good, disentangled, and do not include any bias from the training data. For example, if InsP2P edited results are not good, they can still be edited back to the original image. How does the method avoid this? Besides, which loss contributes most to the model? The table 4 does not clearly show this.

3.	The structure preservation. I think the CLIP similarity loss makes the model have the editing ability. However, this does not indicate the model can preserve the structure since the edited image can be very different from the original but have the same semantics and be edited back to the original. The author may elaborate on this.

4.	Evaluation metrics and results. To validate the editing ability, I think the author should evaluate more diverse editing types and datasets such as PIE.

5.	I am also interested in the attention map visualization in forward and reversed editing. I think the attention can be different at the same timestep since the forward and reverse process are opposite.  Which attention map at which timestep is enforced with consistency? More clarifications and visualization evidence on the loss should be provided.

6.	Besides, the equations are not numbered.

**Questions:**

Please see the weaknesses part.

---

> ### Author Response · Authors · 2024-11-20
> **Rebuttal (1/2)**
>
> > 1. Backpropagation efficiency: The author proposes several consistency losses. Considering the diffusion process InsP2P includes multiple steps to get the final image, what is the backpropagation process of the loss constructed on the image? I think this will be a recursive procedure since the loss will have to backpropagate from the last step of diffusion to the very first step which is insanely computationally expensive and requires large GPU memory. The same questions apply to attention map consistency, etc.
>
> We do not run multiple inference steps while training. Our method, like IP2P and other diffusion models, computes losses based on a single denoising step per iteration, aligned with the pre-defined noise schedule. This ensures all consistency losses, aka Cycle Edit Consistency (CEC), are efficiently calculated without requiring recursive backpropagation through the entire diffusion process. As a result, our approach avoids significant memory overhead and fits comfortably on A100 GPUs (40GB) with a batch size of 512, ensuring computational efficiency. Details about the training are provided in Algorithm 1 (see Sec A.10).
>
> > 2. Validation for removing bias in InsP2P apart from inconsistency. I think the proposed method can maintain the consistency of forward and reversed edits but cannot guarantee that the edited results are good, disentangled, and do not include any bias from the training data. For example, if InsP2P edited results are not good, they can still be edited back to the original image. How does the method avoid this? Besides, which loss contributes most to the model? The table 4 does not clearly show this.
>
> An additional difference between supervised training and our method is that supervised approaches use ground-truth edited images (which often suffer from attribute- or scene-entangled biases, as shown in Fig. 2) as the model's input during training. For IP2P, Algorithm 1 Line 2 is $z_t \gets N(\mathbf{I_{edit}}, t)$, while ours is $z_t \gets N(\mathbf{I_{input}}, t)$. The IP2P procedure introduces biases directly into the model's training. In contrast, our method initializes noise directly from the input image, ensuring that the diffusion process remains grounded in the input's structure and semantics rather than inheriting biases from synthetic edited images, see Sec. A.5 for consecutive application of forward and reverse processes.
>
> Our method incorporates $ \mathcal{L}\_{CLIP} $ and $ \mathcal{L}\_{recon} $ as core components to enforce semantic alignment and reversibility in Cycle Edit Consistency (CEC). Adding $ \mathcal{L}\_{sim} $ allows the model to perform edits more freely, as the base configuration without $ \mathcal{L}\_{sim} $ tends to create outputs similar to the input image. Furthermore, $ \mathcal{L}\_{attn} $ enhances the model's focus on relevant regions and ensures that the region of interest remains consistent between the forward and reverse processes, resulting in better localization and improvements across metrics like L1, L2, CLIP-I, and DINO. We have already provided an ablation study in Tab. 4, demonstrating the individual contributions of loss functions to the overall model performance. To clarify further, we updated the manuscript, see Sec. 5.4 and Sec A.2.
>
> > 3. The structure preservation. I think the CLIP similarity loss makes the model have the editing ability. However, this does not indicate the model can preserve the structure since the edited image can be very different from the original but have the same semantics and be edited back to the original. The author may elaborate on this.
>
> As mentioned in Reviewer aTp8 - Q2, our method starts from a noised version of the input image rather than the ground-truth edited image, ensuring the diffusion process is conditioned on the input and preserving its structure. Structure preservation is also reinforced by $ \mathcal{L}\_{\text{recon}} $, which penalizes deviations in pixel and semantic spaces, and $ \mathcal{L}\_{\text{attn}} $, which aligns attention maps for forward and reverse edits to localize changes to relevant regions. To clarify further, we updated the manuscript; see Sec. 5.4 and Sec A.2. Furthermore, attention maps for the edits have been included in Sec. A.7 to demonstrate the edit localization capabilities of our method. UIP2P focuses on modifying only the relevant parts of the image.

---

> ### Author Response · Authors · 2024-11-20
> **Rebuttal (2/2)**
>
> > 4. Evaluation metrics and results. To validate the editing ability, I think the author should evaluate more diverse editing types and datasets such as PIE.
>
> Thank you for the suggestion. We applied our method to the PIE benchmark to evaluate performance on diverse editing tasks and compared it to IP2P, a representative feed-forward instruction-based editing method and a supervised alternative to our approach. The table below summarizes the results:
>
> | Methods                | Distance ↓ | PSNR ↑ | LPIPS ↓ | MSE ↓ | SSIM ↑ | Whole ↑ | Edit ↑ |
> |-----------------------|------------|---------|----------|--------|---------|---------|--------|
> | **IP2P**             | 57.91      | 20.82   | 158.63   | 227.78 | 76.26   | 23.61   | 21.64  |
> | **Ours**             | 27.05      | 26.85   | 60.57    | 40.07  | 83.69   | 24.78   | 21.89  |
>
> Our method demonstrates significant improvements over IP2P across most metrics, including better preservation of structure (PSNR and SSIM), lower perceptual differences (LPIPS), and reduced mean squared error (MSE). These results validate the scalability and versatility of our approach on a broader benchmark. We included this analysis in the revised manuscript to address your suggestion, see Section A.6.
>
>
> > 5. I am also interested in the attention map visualization in forward and reversed editing. I think the attention can be different at the same timestep since the forward and reverse process are opposite. Which attention map at which timestep is enforced with consistency? More clarifications and visualization evidence on the loss should be provided.
>
> At training time, we sample two different noise steps for the forward and backward processes, which are conditioned on the input image and edit instruction. Attention consistency is enforced between these different noise steps to ensure that the model attends to the same regions during both forward and reverse edits. This is supported by cross-attention scores in instruction-based editing methods, which tend to be more consistent across timesteps, as the edit instruction remains fixed, and the model's focus shifts only to the edited regions (see Sec A.7 in the updated manuscript). Recent works, such as those by Guo, Qin, and Lin, and Simsar et al., show that regularizing attention space with a mask during inference enables localized edits in IP2P. Our method builds on these ideas by incorporating attention regularization into the training phase, making it possible to focus on relevant regions from the start and avoiding the need for additional inference-time modifications. We reference the work by Guo, Qin, and Lin, "Focus on your instruction: Fine-grained and multi-instruction image editing by attention modulation" (CVPR 2024) and Simsar et al., "LIME: Localized image editing via attention regularization in diffusion models" (WACV 2025) to substantiate this claim and are happy to address any further concerns on this topic.
>
> > 6. Besides, the equations are not numbered.
>
> Thank you for pointing this out. We numbered all equations in the revised manuscript for clarity.

---

> > ### Comment · Reviewer_aTp8 · 2024-11-28
> > **Response to the rebuttal**
> >
> > Thanks for the author's efforts in clarification and additional experiments!
> >
> > After reading the rebuttal and other reviewer's comments, I raised my score to 5 for the motivation of the paper, additional experiments, and clarification.
> >
> > There are still limitations and concerns about the proposed pipeline and its validation. First, the whole framework utilized 16 H100 GPU for training (indicated in Line 388) which is not on a reasonable scale considering the data and achievement in this paper. Also, in rebuttal, the author claims the method is launched on an A100 (40GB). I am confused about the training resources used in the paper.
> >
> > Second, I have concerns about the validation of the method, for experiment results, I agree with RW MJQy's comments. The qualitative results do not show a clear advantage. Besides, I have concerns about the logic of the proposed algorithm. Since different timesteps t in the diffusion process can make the image have different semantics and layouts, I think it may not be appropriate to use the intermediate latent and attention maps at different timesteps to enforce consistency. Finally, the paradigm of leveraging CLIP for similarity and consistency training is used in DiffusionCLIP [1] but this point is subtle.
> >
> > 1. DiffusionCLIP: Text-Guided Diffusion Models for Robust Image Manipulation.

---

> > > ### Author Response · Authors · 2024-11-28
> > >
> > > Thank you for taking the time to review our rebuttal and other reviewers’ comments. We appreciate your thoughtful feedback and would like to address the remaining concerns in detail.
> > >
> > > > There are still limitations and concerns about the proposed pipeline and its validation. First, the whole framework utilized 16 H100 GPU for training (indicated in Line 388) which is not on a reasonable scale considering the data and achievement in this paper. Also, in rebuttal, the author claims the method is launched on an A100 (40GB). I am confused about the training resources used in the paper.
> > >
> > > In the main paper, we presented results from our model trained on 16 H100 GPUs (80GB each). However, to address concerns regarding high GPU VRAM requirements, we demonstrated in the rebuttal that our method can also be effectively trained on 16 A100 GPUs (40GB each), accommodating a total batch size of 512.
> > >
> > > For dataset generation, IP2P follows a multi-step process for the dataset generation: (1) creating captions and edit instructions using LLMs, (2) generating input images with Stable Diffusion, and (3) applying Prompt-to-Prompt editing. This workflow requires approximately 1-2 minutes per instance. In contrast, since our method eliminates the need for edited images, it simplifies dataset generation by utilizing LLMs only for generating captions and edit instructions, reducing the time required to just 2-3 seconds per instance. This makes our approach significantly more scalable for dataset creation.
> > >
> > > Additionally, since our method modifies only the training objectives without altering the inference pipeline, the inference time remains identical to that of IP2P.
> > >
> > > > Second, I have concerns about the validation of the method, for experiment results, I agree with RW MJQy's comments. The qualitative results do not show a clear advantage.
> > >
> > > Since the evaluation of results for image editing tasks is subjective, in addition to qualitative results, we provided quantitative evaluations on two benchmarks, MagicBrush and PIE-bench, supported by a comprehensive user study, see Tab. 3. These evaluations highlight the superior performance of our proposed method. Additionally, as requested by RW-MJQy in the Weaknesses section of the response, we have further highlighted the advantages of our approach over existing methods in Sec A.4.
> > >
> > > We also wish to clarify that some images in Fig. 4 are sourced directly from competitor papers to ensure a fair comparison. Specifically, Rows 1, 3, and 10 are from SmartEdit; Rows 2 and 11 are from MagicBrush; and Rows 4-9 are from MGIE. Therefore, these images represent the best results reported in the respective papers, highlighting the competitive performance of our method against state-of-the-art approaches.
> > >
> > > > Besides, I have concerns about the logic of the proposed algorithm. Since different timesteps t in the diffusion process can make the image have different semantics and layouts, I think it may not be appropriate to use the intermediate latent and attention maps at different timesteps to enforce consistency.
> > >
> > > The concern about diffusion timesteps t affecting semantics and layouts is valid for text-to-image models. However, our method leverages input image information to preserve the original semantics and layout while using edit instructions to specify the regions to modify. As shown in Sec. A.7, the diffusion process in our method focuses on the editing region at all timesteps, ensuring consistency and retaining unchanged areas of the input image. By comparing cross-attention maps computed in the forward and reverse processes at different time steps, we implicitly try to enforce this behavior by adding it as a regularization objective in our loss formulation.
> > >
> > > Additionally, we want to clarify that none of our losses are computed on intermediate latents; all losses are calculated directly in the image space. Unlike the original IP2P objective, which operates on noise predictions, our method applies loss functions exclusively in the image space. Please refer to the response to RW-goKN (Q3) for additional details.
> > >
> > > > Finally, the paradigm of leveraging CLIP for similarity and consistency training is used in DiffusionCLIP [1] but this point is subtle.
> > >
> > > Thank you for highlighting DiffusionCLIP. We will include this work in the Related Work section and explicitly outline the differences. While the CLIP loss itself is not novel, as noted in Lines 281-285, our contribution lies in its unique application within a cycle consistency framework. Unlike DiffusionCLIP and StyleGAN-NADA, which utilize this loss for predefined domains or edit types, our approach leverages it to handle a diverse range of edits. By integrating CLIP loss with other loss functions, we enable for the first time training on real images without supervision, marking a significant advancement in instruction-based image editing methods.

---

> > > > ### Author Response · Authors · 2024-12-02
> > > >
> > > > Dear Reviewer aTp8,
> > > >
> > > > Thank you for your thoughtful feedback and updated evaluation. We hope further clarifications align with your expectations.
> > > >
> > > > We are now in the last 24 hours of the rebuttal period for reviewers. If there’s any part of our response or revisions that needs further clarification, please let us know, and we would be happy to address it.
> > > >
> > > > Thank you again for your time and valuable insights.
> > > >
> > > > Kind regards,
> > > >
> > > > Paper 1784 Authors

---

> > > > > ### Author Response · Authors · 2024-12-03
> > > > >
> > > > > Dear Reviewer aTp8, we are thankful for your review. As the rebuttal deadline is coming to an end, please let us know if your concerns are well addressed. We are happy to provide further clarification.

---

### Author Response · Authors · 2024-11-25

**Dear Reviewers and ACs,**

Thank you for taking the time to review our submission. Your thoughtful and constructive feedback has been instrumental in refining our work. We have carefully addressed each concern point by point and incorporated the corresponding changes into the revised manuscript, with updates **highlighted in green** for clarity.

The revisions focus on key areas, including computational efficiency, structure preservation, subjective evaluations, and scalability. Additionally, we have emphasized the practical advantages of our approach, particularly its ability to deliver strong performance without relying on pre-trained supervision or ground-truth edited images, distinguishing it from competing methods.

We hope the revised manuscript meets your expectations and addresses all concerns effectively. **We are happy to address any further concerns.**



Best regards,

Paper 1784 Authors

---

### Meta-Review · Area_Chair_hhyV · 2024-12-17

**Metareview:**

This paper aims to address the limited generalization and the bias of supervised instruction-based image editing. The authors propose a method which does not require paired editing images for training via cycle edit consistency. Multiple types of experiments are conducted to demonstrate the effectiveness of the proposed method. The problem setting and the motivation for removing the triplet are recognized by all reviewers. However, the main cycle consistency idea is similar to CycleGAN and is widely seen, it is not very innovative at the technical level.

Besides, the superiority of the proposed method in the visual quality of generated images is also concerned. The proposed method does not have clear advantages over the compared methods. Although the authors conduct a user study, the setup and results of the study are not very convincing. Firstly, it is not clear why two (instead of one) best results are selected. Secondly, the study has 30 questions, which means only one test sample is edited across 15 image-edit instructions or different test sample are separately edited across only partial image-edit instructions. The sample size is too small. Thirdly, based on the small number of test samples, the similar results (e.g., 22% vs. 20% and 20% vs. 19%) cannot sufficiently demonstrate the clear superiority of the proposed method. Besides, the user study does not report the statistical significance, the validity of the statistical results is uncertain.

In addition, the current user study does not consider the worst case, it cannot reflect whether the image generated by the proposed method have significantly worse quality in some cases. A better approach would be to allow the users to rank all the images generated by the six methods in each question, and compare the six methods after counting the average ranking scores of each method. Based on the above considerations, I think the current manuscript does not match the ICLR’s requirement and I do not recommend to accept this manuscript.

**Additional Comments On Reviewer Discussion:**

The authors provided rebuttals for each reviewer. Reviewer aTp8 provided a response and raise the rating score but still had concerns about the proposed pipeline and its validation. As the other reviewer did not provide a further response, I reviewed this paper and had concerns about the generation quality, the user study and the innovation, which were also raised by reviewer MJQy.

---

### Decision · Program_Chairs · 2025-01-22

Reject